# Effects of *Rosa roxburghii* Tratt Must on the Growth, Nutrient Composition, and Antioxidant Activity of *Pleurotus ostreatus* Mycelia

**DOI:** 10.3390/molecules27113585

**Published:** 2022-06-02

**Authors:** Yinfeng Li, Fei Chen, Xiaozhu Liu

**Affiliations:** Guizhou Institute of Technology, Guiyang 550003, China; liyinfeng@git.edu.cn (Y.L.); feichengit16@163.com (F.C.)

**Keywords:** *Pleurotus ostreatus*, *Rosa roxburghii* Tratt, growth, nutrient composition, antioxidant activity

## Abstract

*Rosa roxburghii* Tratt, a Rosaceae plant endemic to China, produces fruit with high nutritional and medicinal value. The effects of *R*. *roxburghii* must on the growth, nutrient composition, and antioxidant activity of *Pleurotus ostreatus* mycelia was investigated. We measured the mycelial growth rate, proximate composition, amino acid and crude polysaccharide content, and the antioxidant activity of the crude polysaccharides of *P*. *ostreatus* mycelia cultivated under different concentrations of *R*. *roxburghii* must (2%, 4%, and 8%, *v*/*v*). Low concentrations of *R*. *roxburghii* must (2% and 4%) promoted mycelial growth, while a high concentration (8%) inhibited mycelial growth. Low concentrations of *R*. *roxburghii* must had no significant effects on the soluble substances, fat, ash, and crude fiber in *P*. *ostreatus* mycelia, but significantly increased the crude protein and total amino acid contents (*p* < 0.05). The addition of *R*. *roxburghii* must at low concentrations significantly increased the crude polysaccharide content in mycelia (*p* < 0.05) but had no impact on the scavenging of hydroxyl radicals and 2,2-diphenyl-1-picrylhydrazyl (DPPH). Therefore, *R*. *roxburghii* must at low concentration can be used as a substrate for *P*. *ostreatus* cultivation to increase the protein and polysaccharide contents in mycelia.

## 1. Introduction

*Pleurotus ostreatus*, the oyster mushroom, is an edible fungus of the order Agaricales in the class Agaricomycetes and division Basidiomycota [1]. The fruiting bodies of *P*. *ostreatus* are highly favored due to their rich nutrients and pleasant flavor, and *P*. *ostreatus* mushrooms are widely consumed. *P*. *ostreatus* is the third most consumed edible mushroom in the world following *Lentinus edodes* and *Agaricus bisporus*, and the most abundantly and widely cultivated edible mushroom in China [2,3]. With the rapid scaling-up of *P*. *ostreatus* cultivation, the shortage of substrates and the increase in production costs have become a bottleneck in the development of the *P*. *ostreatus* industry. Therefore, there is a pressing need to identify alternative culture substrates for the production of *P*. *ostreatus*.

*Rosa roxburghii* Tratt is a small deciduous perennial shrub in the Rosaceae family. This species is also known as “Cili” because the fruit is covered with prickles [4,5]. The *R*. *roxburghii* endemic to China is mainly distributed in the southwestern regions, such as Guizhou, Yunnan, and Sichuan provinces [6]. Studies have shown that both the pomace and must of *R*. *roxburghii* are rich in minerals, organic acids, and amino acids [7]. These nutrients are sufficient for the growth of edible mushrooms, and *R*. *roxburghii* fruit pomace and must can be used as excellent cultivation substrates. Yang et al. [8] demonstrated that *R*. *roxburghii* fruit pomace as a nitrogen source could support the growth of *P*. *ostreatus* by measuring the protein content during the cultivation of *P*. *ostreatus* with *R*. *roxburghii* pomace. Zhang et al. [9] reported the feasibility of *P*. *ostreatus* cultivation with *R*. *roxburghii* pomace. Yang et al. [10] further optimized the formulation of *R*. *roxburghii* pomace as a substrate for *P*. *ostreatus* production. Therefore, it is feasible to use *R*. *roxburghii* pomace for the cultivation of *P*. *ostreatus*. The use of this substrate not only leads to rapid fruiting body formation and mycelial growth, but also reduces the environmental pollution caused by pomace. Whether the must can also be used as a substrate for *P*. *ostreatus* cultivation, and whether using the must would affect the nutrient composition of *P*. *ostreatus* mycelia are questions that require further investigation.

In this study, the effects of different levels of *R*. *roxburghii* must on the growth rate of *P*. *ostreatus* mycelia were determined. Then, the proximate composition and the contents of amino acids and crude polysaccharides in the mycelia of *P*. *ostreatus* cultivated with must were examined. Finally, the antioxidant activity of the crude polysaccharides in the mycelia cultivated with fruit juice was analyzed.

## 2. Materials and Methods

### 2.1. Sources of Materials

*P. ostreatus* of the cultivar P10 was purchased from the China Center of Industrial Culture Collection and maintained in a potato dextrose agar medium (PDA: 200 g/L potato infusion, 20 g/L dextrose, and 15 g/L agar) at 4 °C for later usage.

The *R. roxburghii* fruit of the cultivar Guinong 5 was obtained from Longli County, Guizhou Province, China (Figure 1). Fresh and mature *R. roxburghii* fruits were washed with sterile water, crushed with a juice extractor, and filtered through microporous film (0.22 μm) to acquire the must, which was then added to the medium for the cultivation of the mycelium of *P. ostreatus*.

### 2.2. Effect of R. roxburghii Must on the Growth of P. ostreatus Mycelia

First, PDA containing *R. roxburghii* must (RM) with a concentration of 3% (*v*/*v*) was used to evaluate whether *R. roxburghii* must would affect the growth of mycelia. The PDA without *R. roxburghii* must was set as a control. The mycelium discs (1 cm in diameter) of *P. ostreatus* were placed in PDA medium plates under aseptic conditions and incubated at 28 °C in darkness for 5 days. The diameter of the mycelium was measured, and the colony morphology was noted [11]. Three parallel replicates were performed for each sample.

Then, three different additive amounts of *R. roxburghii* must with concentrations of 2%, 4%, and 8% (*v*/*v*) were added to the PDA. The diameters of the three groups of mycelia were recorded and compared with the control group using the above method.

Cottonseed hull medium was prepared at a substrate-to-water ratio of 1:1.2 (containing 1% lime) and sterilized in a test tube (20 cm × 200 mm) at 121 °C for 30 min. Identical mycelial plugs of *P*. *ostreatus* were made with a punch (1 cm in diameter), inoculated into the test tubes with the cottonseed hull medium, and cultured at 28 °C. The mycelial growth was observed and recorded. Each sample had three replicates.

### 2.3. Effects of R. roxburghii Must on the Nutrient Composition of P. ostreatus Mycelia

The mycelia of *P*. *ostreatus* cultured in each group were dried in an oven to a constant weight and ground into powder for determination of the nutrient composition.

Determination of the proximate composition: crude protein was measured using the micro-Kjeldahl method [12]; the fat content was measured by Soxhlet extraction [13]; and the measurement of soluble substances, ash, and crude fiber was performed following the methods described by Wang et al. [14].

Determination of amino acids: dry mycelia were hydrolyzed with hydrochloric acid (6 mol/L), and the amino acid content was determined using an automatic amino acid analyzer (A300, MembraPure, Hennigsdorf, Germany). Cysteine and methionine were measured as cysteic acid and methionine sulfone, respectively, after performing acid oxidation and 20% HCl hydrolysis at 150 °C for 20 h. The tryptophan analysis was conducted on a Ba(OH)_2_ hydrolysate.

### 2.4. Effects of R. roxburghii Must on the Crude Polysaccharide Content and Antioxidant Activity of P. ostreatus Mycelia

One hundred grams of dry powder from the *P. ostreatus* mycelia was prepared and the crude polysaccharides extracted with hot water (65 °C, 2.5 h, 1:30, *w*/*w*). The extract was filtered and concentrated at 65 °C in a rotary evaporator under reduced pressure, precipitated with 95% ethanol at a 1:3 ratio (extract:ethanol, *v*/*v*), and kept overnight at 4 °C. The extract was then centrifuged (4000× *g*, 30 min). The obtained precipitate was freeze-dried, and the crude polysaccharide preparation was finished. Finally, the yield and content of crude polysaccharides were calculated.

Analysis of antioxidant activity: the assessment of the scavenging ability of the crude polysaccharides of *P. ostreatus* on hydroxyl radicals and 2,2-diphenyl-1-picrylhydrazyl (DPPH) radicals was conducted according to the method previously described by Zhang [15].

### 2.5. Statistical Analysis

The results were the mean ± standard deviation of a triplicate analysis. Statistical comparisons were conducted using the Student’s *t* test. Differences were considered statistically significant at *p* < 0.05.

## 3. Results and Analysis

### 3.1. Physicochemical Parameters of R. roxburghii Must

The cultivar of *R*. *roxburghii* used in this study was Guinong 5 (Figure 1), which was developed in Longli County, Guizhou Province, China, with an average fruit weight of 21.26 ± 1.58 g. The total sugar content of the fruit juice was 5.22 ± 0.34 g/100 g FW, the total acid content was 987.13 ± 73.69 mg/100 g FW, and the soluble protein content was 0.34 ± 0.02 g/100 g FW (Table 1).

### 3.2. Effects of R. roxburghii Must on the Growth of P. ostreatus Mycelia

To analyze the effects of *R*. *roxburghii* must on the growth of *P*. *ostreatus* mycelia, the growth of mycelia on the PDA medium with (3% RM *v*/*v*) and without (control) must were compared. The results showed that the mycelia were denser and thicker, and the growth was faster with must (Figure 2A,B). The mycelial diameter and growth rate with must were significantly improved compared to the control (Figure 2C,D).

Further analysis focused on whether the effects of must on mycelial growth were concentration-dependent. *R*. *roxburghii* must was added to the PDA medium at different concentrations, and the mycelial diameter and growth rate were measured. The results showed that the mycelial diameter and growth rate significantly increased with increasing *R*. *roxburghii* must concentrations in the range of 0 to 4%. Mycelial diameter and growth rate reached the maximum at 4% RM but decreased at 8% RM (Figure 3).

In addition, this study analyzed the growth characteristics of *P*. *ostreatus* mycelia on a cottonseed hull medium. The mycelial growth on the cottonseed hull medium with different concentrations of *R*. *roxburghii* must was similar to that in the PDA medium. Mycelial growth was significantly accelerated by the addition of 2% and 4% RM but was inhibited by 8% fruit juice (Figure 4). The inhibition of mycelial growth with 8% RM in the cottonseed hull medium was less than that in the PDA medium.

The results showed that low concentrations of *R*. *roxburghii* must promoted the growth of *P*. *ostreatus* mycelia, while high concentrations of must inhibited mycelial growth. The optimum concentration of 4% was used for the subsequent experiments.

### 3.3. Effects of R. roxburghii Must on the Nutrient Composition of P. ostreatus Mycelia

To investigate the effects of *R*. *roxburghii* must on the nutrient composition of *P*. *ostreatus* mycelia, the differences in the proximate composition, amino acid composition, and vitamin content in mycelia with (4%) and without (control) fruit juice were analyzed.

Table 2 shows the proximate composition of *P*. *ostreatus* mycelia. There were no significant differences in the soluble substances, fat, ash, or crude fiber content in mycelia with and without *R*. *roxburghii* must. However, the content of crude protein was significantly increased with the addition of *R*. *roxburghii* must.

Table 3 displays the differences in amino acids in the mycelia of *P*. *ostreatus*. The total amino acid content in the mycelia cultured with *R*. *roxburghii* must (12.29 ± 0.41%) was significantly higher than that in the control (11.23 ± 0.37%), though there was no significant difference in the total content of essential amino acids (5.02 ± 0.18% with RM vs. 5.42 ± 0.17% without). The content of phenylalanine was significantly reduced by the addition of *R*. *roxburghii* must, whereas the levels of eight amino acids (threonine, alanine, isoleucine, lysine, proline, tryptophan, arginine, and glutamic acid) were significantly increased.

An analysis of the content of vitamins in the mycelia of *P*. *ostreatus* cultured with and without *R*. *roxburghii* must was also conducted. As shown in Table 4, no significant differences in thiamine, riboflavin, and niacin were observed.

### 3.4. Effects of R. roxburghii Must on the Polysaccharide Content and Antioxidant Activity of P. ostreatus Mycelia

The results showed that there was no significant difference in the yield of crude polysaccharides between the two groups, but *R*. *roxburghii* must significantly increased the crude polysaccharide content in the *P*. *ostreatus* mycelia (Figure 5).

Further analysis was conducted on the antioxidant activity of the crude polysaccharides in the mycelia of *P*. *ostreatus* cultivated with and without *R*. *roxburghii* must. As shown in Figure 6A, the scavenging of hydroxyl radicals increased with the increase in polysaccharide concentration. There was no significant difference in the scavenging of hydroxyl radicals by crude polysaccharides between the two groups. Figure 6B illustrates the scavenging of DPPH by the mycelial polysaccharides of *P*. *ostreatus*. The scavenging of DPPH in both groups increased with increasing crude polysaccharide concentration, and no significant difference was found between the two groups.

## 4. Discussion

*P*. *ostreatus* is an edible mushroom that is widely cultivated around the world because of its unique flavor and high adaptability to cultivation substrates [16]. With the worldwide grain shortage and rising grain prices, especially due to the impact of the COVID-19 pandemic on agricultural production in the past two years, the cultivation of *P*. *ostreatus* has been greatly impacted. Therefore, it is necessary to find alternative substrates for *P*. *ostreatus* production. As a by-product of the fruit processing industry, pomace can be used as a substrate for *P*. *ostreatus* cultivation and has the advantages of low cost, easy accessibility, and low technical complexity [17]. Apple [18], grape [19], and grapefruit [20] pomaces have been studied and confirmed as high-quality substrates for the cultivation of *P*. *ostreatus*.

*R*. *roxburghii* is endemic to the southern regions of China and has high nutritional and medicinal value. The fruit has received extensive attention and intensive investigation in recent years [21], and the planting area of *R*. *roxburghii* is increasing every year. In Guizhou Province, for example, the planting area reached 140,000 hectares in 2021, and the total industrial output value exceeded USD 500 million, an increase of more than 53% over the same period last year. Therefore, China is rich in *R*. *roxburghii* resources. Unfortunately, the fruit contains a high phenol content, giving it a sour, astringent, and unpleasant taste, and is, therefore, often used for deep processing to produce juice and pomace. The pomace has been shown to be useful in the cultivation of *P*. *ostreatus*, with the advantages of fast fruiting body formation, a short growth cycle, and high production efficiency [8,9,10]. Whether the must of *R*. *roxburghii* can also be used as a substrate for the production of *P*. *ostreatus* has remained unknown. The results of this study revealed that low concentrations of *R*. *roxburghii* must (≤4% *v*/*v*) promoted the growth of *P*. *ostreatus* mycelia, while a high concentration (8% *v*/*v*) inhibited mycelial growth. Therefore, *R*. *roxburghii* must at low concentrations can be used as a substrate for the cultivation of *P*. *ostreatus* mycelia.

Studies have shown that the growth of *P*. *ostreatus* mycelia can be influenced by multiple factors [22]. For example, a high histidine content in media promotes mycelial growth in *P*. *ostreatus* [23], while trace elements can also affect the growth of mycelia. In a previous study, Zn^2+^ promoted branching and increased the dry weight of mycelia, while Mn^2+^ and Cu^2+^ inhibited the elongation of *P*. *ostreatus* mycelia [24]. The carbon-to-nitrogen ratio in media can also affect the growth rate of *P*. *ostreatus* mycelia [25]. The must of *R*. *roxburghii* contains a variety of trace elements such as iron, zinc, and copper, in addition to abundant amino acids and vitamins [26]. These nutrients may have various impacts on the mycelial growth of *P*. *ostreatus*. Therefore, this study demonstrated that the growth of *P*. *ostreatus* mycelia increased and then decreased with increasing concentrations of *R*. *roxburghii* must. The underlying reasons for this finding need to be further studied.

The results also showed that must at low concentration (4% *v*/*v*) significantly increased the crude protein and amino acid contents in the mycelia of *P*. *ostreatus*. This may be explained by the rich carbon and nitrogen sources in *R*. *roxburghii* must.

Polysaccharides have been extracted from the fruiting bodies, mycelia, and mycelial fermentation broth of *P*. *ostreatus*, and exhibit reported antioxidant, antitumor, and antiviral activities [27,28]. In the present study, addition *R*. *roxburghii* must could increase the content rather than the yield of crude polysaccharides in mycelia. The antioxidant activities of mycelial polysaccharides towards hydroxyl radicals and DPPH were similar under both culture conditions.

## 5. Conclusions

*R*. *roxburghii* must improved the overall antioxidant performance of polysaccharides in the mycelia of *P*. *ostreatus*. *R*. *roxburghii* must had dual effects on the growth of *P*. *ostreatus* mycelia: low concentrations (not more than 4% *v*/*v*) promoted the growth of *P*. *ostreatus* mycelia, while a high concentration (8% *v*/*v*) inhibited the growth of mycelia. The addition of *R*. *roxburghii* must had no effect on the soluble substances, fat, ash, and crude fiber in the mycelia of *P*. *ostreatus*, but significantly increased the crude protein and amino acid contents. In addition, *R*. *roxburghii* must increased the content of crude polysaccharides in the mycelia and thus improved the total antioxidant activity. Taken together, these findings indicate that *R*. *roxburghii* must, at appropriate concentrations, can be used as a natural substrate for the cultivation of *P*. *ostreatus* mycelia.

## Figures and Tables

**Figure 1 molecules-27-03585-f001:**
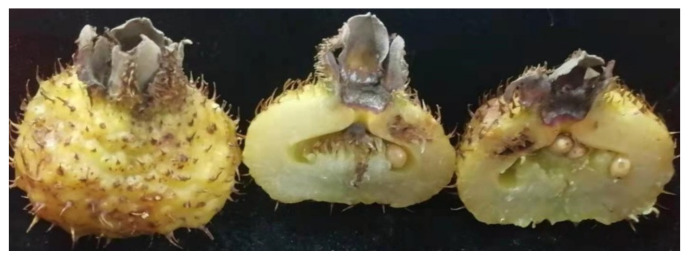
External and internal morphology of “Guinong 5” *Rosa roxburghii* Tratt fruit used in this study.

**Figure 2 molecules-27-03585-f002:**
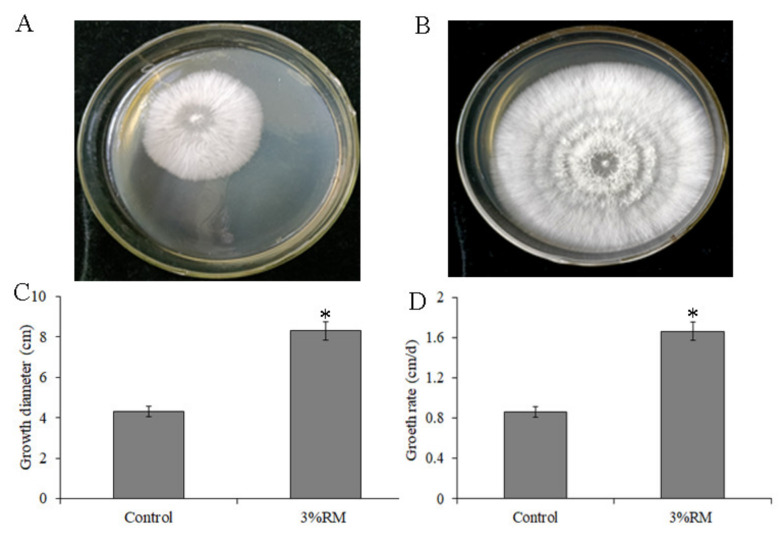
*R*. *roxburghii* must (RM) promoted the growth of *P**. ostreatus* mycelia. (**A**): control group (without must); (**B**): must group (3% *v*/*v*); (**C**): diameter of *P*. *ostreatus* mycelia cultured for 5 days on a potato dextrose agar (PDA) medium contain 3% *R*. *roxburghii* must or not; (**D**): growth rate of *P*. *ostreatus* mycelia on a PDA medium contain 3% *R*. *roxburghii* must or not. * indicates a significant difference at *p* < 0.05 compared with the control group.

**Figure 3 molecules-27-03585-f003:**
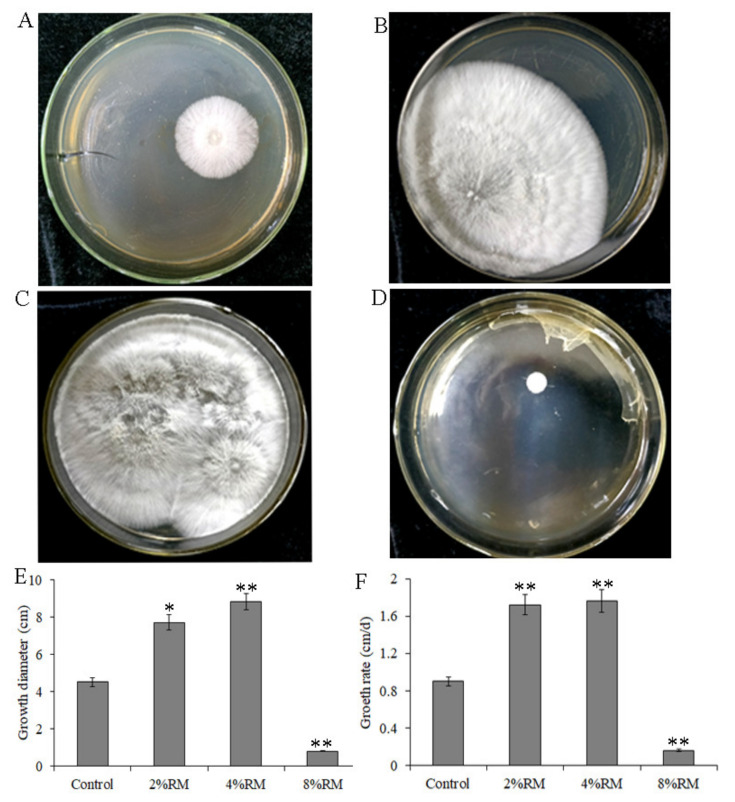
Growth of *P**. ostreatus* mycelia on a potato dextrose agar (PDA) medium with different concentrations of *R*. *roxburghii* must (RM). (**A**): control group (without must); (**B**): 2% RM (*v*/*v*); (**C**): 4% RM (*v*/*v*); (**D**), 8% RM (*v*/*v*); (**E**): diameter of mycelia on the PDA medium with different concentrations of *R*. *roxburghii* must; (**F**): growth rate of mycelia on the PDA medium with different concentrations of *R*. *roxburghii* must. * indicates a significant difference at *p* < 0.05 compared with the control group; ** indicates a significant difference at *p* < 0.01 compared with the control group.

**Figure 4 molecules-27-03585-f004:**
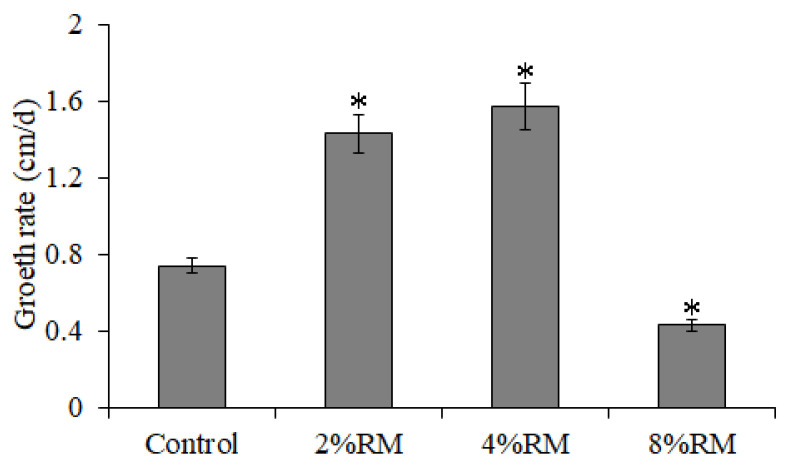
Growth rate of *P**. ostreatus* mycelia on a cottonseed hull medium with different concentrations of *R*. *roxburghii* must. * indicates a significant difference at *p* < 0.05 compared with the control group.

**Figure 5 molecules-27-03585-f005:**
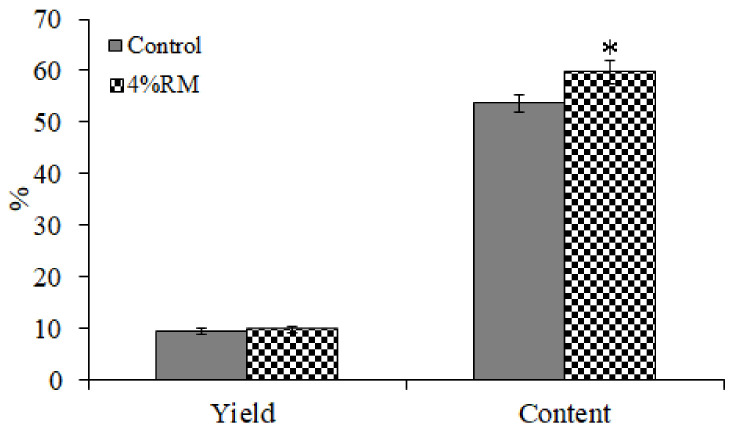
Yields and contents of crude polysaccharides extracted from the *P. ostreatus* mycelium grown on potato dextrose agar (PDA) (%, dry weight basis). * indicates a significant difference at *p* < 0.05 compared with the control group.

**Figure 6 molecules-27-03585-f006:**
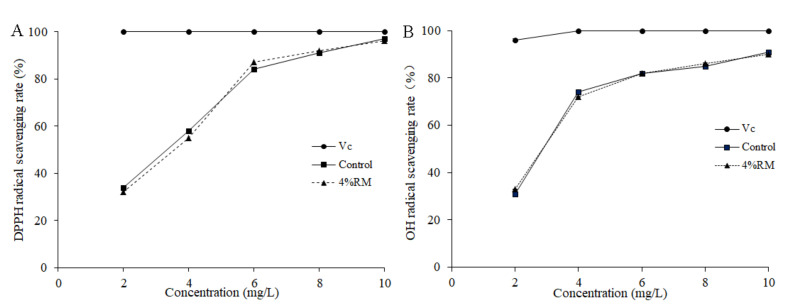
Antioxidant activity of polysaccharides extracted from the *P. ostreatus* mycelium. (**A**): hydroxyl (OH) radical scavenging rate; (**B**): 2,2-diphenyl-1-picrylhydrazyl (DPPH) radical scavenging rate. Vc: vitamin C.

**Table 1 molecules-27-03585-t001:** Physicochemical properties of *R*. *roxburghii* fruit and must.

	Weight (g)	pH	Content (g/100 g FW)		
			Total Sugar	Total Acid	Soluble Protein
Fruit	21.26 ± 1.58	-	-	-	-
Must	-	3.49 ± 0.13	5.22 ± 0.34	9.87 ± 0.73	0.34 ± 0.02

**Table 2 molecules-27-03585-t002:** Proximate composition of the *P**. ostreatus* mycelium grown on potato dextrose agar (%, dry weight basis). * indicates a significant difference at *p* < 0.05 compared with the control group.

Groups	Crude Protein	Soluble Substances	Fat	Ash	Crude Fiber
Control	15.36 ± 0.43	13.25 ± 0.71	3.51 ± 0.14	5.68 ± 0.32	15.36 ± 0.73
4% RM	18.97 ± 0.56 *	14.13 ± 0.64	3.54 ± 0.26	6.03 ± 0.28	15.41 ± 0.66

**Table 3 molecules-27-03585-t003:** Amino acid composition of the *P**. ostreatus* mycelium grown on potato dextrose agar (%, dry weight basis). * indicates a significant difference at *p* < 0.05 compared with the control group. ^a^ The essential amino acids.

Amino Acids	Groups	
	Control	4% RM
Aspartic acid	1.35 ± 0.04	1.41 ± 0.05
Threonine ^a^	0.51 ± 0.02	0.65 ± 0.02 *
Valine ^a^	0.82 ± 0.02	0.84 ± 0.04
Glycine	0.52 ± 0.01	0.51 ± 0.02
Serine	0.46 ± 0.02	0.43 ± 0.01
Alanine	0.66 ± 0.03	0.85 ± 0.03 *
Cysteine	0.10 ± 0.00	0.08 ± 0.01
Leucine ^a^	0.94 ± 0.03	0.92 ± 0.01
Isoleucine ^a^	0.63 ± 0.04	0.75 ± 0.02 *
Methionine ^a^	0.53 ± 0.02	0.58 ± 0.02
Tyrosine	0.28 ± 0.01	0.33 ± 0.01
Phenylalanine ^a^	0.82 ± 0.03	0.68 ± 0.02 *
Histidine	0.18 ± 0.01	0.14 ± 0.01
Lysine ^a^	0.42 ± 0.00	0.54 ± 0.02 *
Proline	0.53 ± 0.02	0.62 ± 0.03 *
Tryptophan ^a^	0.35 ± 0.02	0.46 ± 0.02 *
Arginine	0.61 ± 0.02	0.76 ± 0.03 *
Glutamic acid	1.52 ± 0.03	1.74 ± 0.05 *
Total essential amino acids	5.02 ± 0.18	5.42 ± 0.17
Total amino acids	11.23 ± 0.37	12.29 ± 0.41 *

**Table 4 molecules-27-03585-t004:** Vitamin content of the *P. ostreatus* mycelium grown on PDA (mg/100 g, dry weight basis).

Groups	Thiamine	Riboflavin	Niacin
Control	1.96 ± 0.12	3.41 ± 0.18	94.56 ± 4.35
4% RM	1.94 ± 0.15	3.54 ± 0.26	101.79 ± 6.48

## Data Availability

Not applicable.

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
