# Peer review of "Effects of Rosa roxburghii Tratt Must on the Growth, Nutrient Composition, and Antioxidant Activity of Pleurotus ostreatus Mycelia"

_molecules, 2022, doi:10.3390/molecules27113585_

Round 1

Reviewer 1 Report

Report on the manuscript molecules-1755208

The manuscript submitted by X. Liu & al. entitled « Effects of Rosa roxburghii Tratt must on the growth, nutrient composition, and antioxidant activity of Pleurotus ostreatus mycelia », describes the effects of R. roxburghii must on the growth, nutrient composition, and antioxidant activity of Pleurotus ostreatus mycelia. The conclusion is that the addition of R. roxburghii must at low concentrations significantly increased crude polysaccharide content in mycelia (P<0.05), but had no impact on the scavenging of hydroxyl radicals and 2,2-diphenyl-1-picrylhydrazyl (DPPH). Consequently, R. roxburghii must at low concentration can be used as substrate for P. ostreatus cultivation to increase the protein and polysaccharide contents in mycelia.

 This study is interesting and well conducted, which includes the description of the experimental conditions.

The text is short but precise, and sheds a clear light on the clearly positive effects of this contribution to the growth of the mycelia.

These results will have an impact on the increase in productivity and the costof production of P. ostreatus.

Some minor corrections would have to be made, in particular at the level of references.

Ref 1 : Last name nust appear instead of first name : Mbassi (Josiane), Mobou (Estelle), Ajebesone (Francis Ngome), Sado (Kandem)

Ref 11 : Pak. J. Biol. Sci.

Ref 12 : J. Radioanal. Nucl. Chem.

Ref 24 : Pleurotus ostreatus hyphal growth

Author Response

Dear Editor:

This is a revision of our previous manuscript (molecules-1755208). First, we thank the reviewers for their critical comments and constructive suggestions. Based on the suggestions, we performed many revisions of this manuscript. For the response to reviewers, the comments are black, author's answers are blue.

Sincerely yours,

Professor Xiaozhu Liu

Guizhou Institute of Technology, Guiyang, China

Reviewer 2 Report

 Major comments

1. It is highly important to make a comparison of R.r.T must effect on P.ostreatus mycelia growth with other known growth stimulators using either own data, or those published by other authors

2. Why two medium: cotton hull and PDA?- no explanation is given

2. ‘high histidine content in media promotes mycelial growth’- then why histidine has not been determined?

3. As growth of mycelium depends on minerals supply why didn’t the authors analyze minerals concentrations?

4. Conclusion needs revision ‘The addition of R. roxburghii must have no effects on soluble substances, fat, ash, and crude fiber in the mycelia of P. ostreatus, but significantly increased the crude protein and amino acid contents.”- is it so? See Table 3 data: there are no differences in amino acids content except threonine

5.Abstract needs revision ‘Low concentrations of R. roxburghii must had no significant effects on soluble substances, fat, ash, and crude fiber in P. ostreatus mycelia, but significantly increased the crude protein and total amino acid contents (P<0.05).’- there are no data on the total amino acids content in the text

Minor comments

  1. The authors seldom attach the text of the manuscript to the presented Figures. For instance, Figure 3 citation is absent in the text
  2.  sometimes there are mistakes in Tables citation: ‘As shown in Table 3, no significant differences in thiamine, riboflavin, and niacin were observed’.-table 3 deals only with aminoacids
  3. Delete repetition: ‘However, the content of crude protein was significantly increased with the addition of R. roxburghii must. Therefore, the addition of R. roxburghii must could increase the protein content in the mycelia of P. ostreatus’
  4. Fig.6- what is Vc?
  5. Change ‘3. RESULTS AND ANALYSIS’ to ‘3. RESULTS AND DISCUSSION’
  6. Material and methods section: ‘fat content was measured by Soxhlet extraction [13]; and the measurement of soluble substances, ash, and crude fiber was performed following the methods described by Wang et al. [14]’. ‘- it is necessary to give additional details of analysis what are they based on? If you speak about fat, then write what solvent was used. No data about soluble compounds, crude fiber. I have tried to find the details in the citation work, but failed. While reading the manuscript a person should understand what methods were used in the work without looking through Internet data.
  7. Reference list- should be revised according to the authors rules (use bold letters for year of publications and italics for the title of the journal and volume)
  8. ‘Determination of amino acids: dry mycelia were hydrolyzed with acid’- what acid???

Author Response

(The authors gave the same response as above.)
